# Structural and Socio-Spatial Determinants Influencing Care and Survival of Patients with a Pancreatic Adenocarcinoma: Results of the PANDAURA Cohort

**DOI:** 10.3390/cancers14215413

**Published:** 2022-11-03

**Authors:** Gaël S. Roth, Yohan Fayet, Sakina Benmameche-Medjahed, Françoise Ducimetière, Amandine Charreton, Claire Cropet, Sylvie Chabaud, Anne-Marie Marion-Audibert, Olivier Berthelet, Thomas Walter, Mustapha Adham, Mathieu Baconnier, David Tavan, Nicolas Williet, Pascal Artru, Floriane Huet-Penz, Isabelle Ray-Coquard, Fadila Farsi, Hélène Labrosse, Christelle de la Fouchardière

**Affiliations:** 1University Grenoble Alpes/Hepato-Gastroenterology and Digestive Oncology Department, CHU Grenoble Alpes/Institute for Advanced Biosciences, CNRS UMR 5309-INSERM U1209, 38043 Grenoble, France; 2Research on Healthcare Performance (RESHAPE), INSERM U1290, Equipe EMS-Social and Human Sciences Department, Centre Léon Bérard, 69008 Lyon, France; 3Medical Oncology Department, Centre Léon Bérard/Research Center of Lyon (CRCL), UMR INSERM 1052, 28 rue Laennec, 69008 Lyon, France; 4Biostatistics Deparment, Centre Léon Bérard/Research Center of Lyon (CRCL), UMR INSERM 1052, 28 rue Laennec, 69008 Lyon, France; 5Gastroenterology and Hepatology, Clinique du Val d’Ouest, 39 Chemin de la Vernique, 69130 Ecully, France; 6Hepato-Gastroenterology Department, Centre Hospitalier Métropole Savoie, 73011 Chambery, France; 7Medical Oncology Department, Hopital Edouard Herriot, 69003 Lyon, France; 8Surgery Department, Hopital Edouard Herriot, 69003 Lyon, France; 9Hépatogastroenterology Department, Centre Hospitalier Annecy Genevois, 74370 Epagny Metz-Tessy, France; 10Hépatogastroentérology Department, Infirmerie Protestante, 69300 Caluire-et-Cuire, France; 11Hepatogastroenterology and Digestive oncology Department, University Hospital of Saint-Etienne, 42270 Saint-Priest-en-Jarez, France; 12Gastroenterology Department, Hopital privé Jean Mermoz, 69008 Lyon, France; 13Hepato-Gastroenterologie, Centre Hospitalier Alpes Léman, 74130 Contamine sur Arve, France; 14ONCOAURA, Dispositif Spécifique Régional de Cancérologie, 60 Avenue Rockefeller, 69008 Lyon, France

**Keywords:** pancreatic adenocarcinoma, socio-spatial disparities, care delays, general practitioner, localized potential accessibility, pancreatic surgery

## Abstract

**Simple Summary:**

Pancreatic cancer is often diagnosed at an advanced stage, complicated to manage, and highly lethal due to its aggressivity. Optimizing its diagnosis and care pathways is a key challenge. This multicentric French cohort included 538 patients with pancreatic adenocarcinoma in 76 French centers. Among the most important results, the delays of care did not statistically influence survival in this cohort; high access to a general practitioner was associated with better chances to be resectable and survival was correlated with volume of pancreatic surgeries in healthcare centers.

**Abstract:**

Background and aims: Pancreatic cancer is highly lethal and often diagnosed at an advanced stage. This cohort study analyzes the impact of care pathways, delays, and socio-spatial determinants on pancreatic cancer patients’ diagnosis, treatment, and prognosis. Method: Patients with pancreatic adenocarcinoma newly diagnosed at all stages between January and June 2016 in the AuRA French region were included. The influence on survival of delays of care, healthcare centers’ expertise, and socio-spatial determinants was evaluated. Results: Here, 538 patients were included in 76 centers including 116 patients (21.8%) with resectable, 64 (12.0%) borderline-resectable, 147 (27.6%) locally-advanced tumors, and 205 (38.5%) with metastatic disease. A delay between first symptoms and CT scans did not statistically influence overall survival (OS). In resected patients, OS was significantly higher in centers with more than 20 surgeries (HR_<5 surgeries/year_ = 2.236 and HR_5-20 surgeries/year_ = 1.215 versus centers with > 20 surgeries/year *p* = 0.0081). Regarding socio-spatial determinants, patients living in municipalities with greater access to a general practitioner (HR = 1.673, *p* = 0.0153) or with a population density below 795.1 people/km^2^ (HR = 1.881, *p* = 0.0057) were significantly more often resectable. Conclusion: This cohort study supports the pivotal role of general practitioner in cancer care and the importance of the centralization of pancreatic surgery to optimize pancreatic cancer patients’ care and outcomes. However, delays of care did not impact patient survival.

## 1. Introduction

Pancreatic ductal adenocarcinoma (PDAC) represents a worldwide medical challenge with a five-year overall survival (OS) under 10% due to late diagnosis, tumor aggressivity, and a very limited therapeutic arsenal [1,2]. PDAC’s incidence is increasing and statistical projections position it as being the second cause of cancer-related death in Western countries by 2030 [3,4]. Even though the relapse rate is high, surgery is the only intent-to-cure treatment to propose to these patients. Resection requires a very early diagnosis and is possible in only 10–20% of cases at the diagnosis [1]. The common sense hypothesizing that an early recognition and a short delay before starting an appropriate treatment should theoretically lead to a better prognosis in PDAC has been intuitively accepted as true. However, to what extent waiting times and, in particular, the diagnosis delay, have an impact on the OS of PDAC patients remains controversial due to opposite results in the literature [5,6,7]. Moreover, the idea that multiple factors such as socio-spatial inequalities in delay and in quality of care may interfere with cancer treatment and prognosis is growing in the literature [8]. Indeed, several studies reported disparities in access to care between PDAC patients and how it may have an impact on their prognosis [9,10,11]. Thus, the French regional Pancreas Program PANDAURA, launched in 2019, aimed to analyze the care pathways of PDAC patients and provide useful knowledge to improve them and reduce potential inequalities in Auvergne–Rhône–Alpes (AuRA), which is a large region (69,711 km^2^) of 8,000,000 inhabitants (approximately 12% of the French population) with very heterogenous territories such as metropoles, agricultural areas, or mountains. This project brought together various key healthcare players in AuRA as the Regional Health Agency, the Regional Cancer Network, some regional unions of health care practitioners (GP, nurses, and specialists), as well as a broad spectrum of clinical centers such as Universitary Hospitals, Comprehensive Cancer Centers, and health care facilities from public (Hospital) and private (Clinics) sectors. Preliminary studies using data from the national hospital discharge summaries database system pointed out the inverse correlation between the level of expertise in the health facility in which the patient had their first hospital stay and the likelihood of undergoing any specific treatment for PDAC [12]. This result also suggested the potential impact of the level of expertise of healthcare centers on the quality of care in complex pathologies such as PDAC.

Following these preliminary results, the regional multicentric PANDAURA cohort was built to study the care pathways of PDAC patients in depth, considering healthcare actors, diagnostic and therapeutic strategies’ organization, and care delays, as well as potential socio-spatial inequalities and their impact on patient prognosis and survival.

## 2. Patients and Method

### 2.1. Study Objective

The objective of this study was to identify clinical, structural, and socio-spatial determinants impacting care and survival of patients with pancreatic adenocarcinoma.

### 2.2. Population

The inclusion criteria were as follows: any patient over 18 years old diagnosed between January and June 2016 in the AuRA French region for a PDAC, with no prior antitumor treatment, at all stages of disease (resectable, unresectable, and metastatic, according to the National Comprehensive Cancer Network criteria). Relapses of pancreatic cancer and other histological diagnoses were excluded. The identification of PDAC cases was based on pathology reports provided by the 33 pathology departments of the AuRa region and based on patient files presented during multidisciplinary concertation meetings, as routinely done in France for cancer patients. Then, useful clinical information was collected by a clinical research associate in 76 facilities of the AuRA region.

### 2.3. Ethical Approval

The study complied with ethical standards and the Helsinki Declaration of 1975, as revised in 2008, and was authorized on 8 August 2019 by the French Data Protection Authority “Commission Nationale de l’Informatique et des Libertés” (CNIL)-N°919240.

### 2.4. Socio-Spatial Indices

Using patients’ municipality of residence at diagnosis, municipality-level socio-spatial indices were incorporated to investigate potential inequalities in PDAC management and survival as follow: population density; the French deprivation index (Fdep) built from a set of four variables (income, education, unemployment, and working-class jobs); the nationwide Localized Potential Accessibility (LPA) index, which measures spatial accessibility to general practitioners (GP) in France using the two-step floating catchment area method [13]; the spatial accessibility to a specialized hospital, estimated with Odomatrix software based on the average road travel time by car to the nearest University Hospital or Comprehensive Cancer Center; the Geographic Classification for Health Studies (GeoClasH) built from 10 variables measuring the physical, social, and medical environments [14]. Patients were divided into quintiles for each continuous variable.

### 2.5. Statistics

The patient and disease characteristics were described using count and frequencies for qualitative variables, and median with interquartile range for quantitative variables. Socio-spatial items were analyzed using quartiles.

OS was defined as the time from diagnosis to the date of death or date of last news. The OS was estimated using the Kaplan–Meier method. A Cox model multivariate analysis was performed in order to assess the impact of clinical, environmental, and social determinants on OS. The following variables were included in the initial full model on the global population: age, sex, body mass index (BMI), circumstances of diagnosis, history of diabetes, disease stage, type of care center, and socio-spatial items. When performing the multivariate analysis on resected patients the following items were added: resection margin status, surgery-related adverse events, and number of pancreatic surgeries per center per year. When performing the model on patients treated with surgery and/or chemotherapy and/or radiotherapy, the model was the same as the full model on the global cohort, excluding socio-spatial items and type of care center, and adding the duration between CT scan and first treatment. For each model, a backward procedure was used to keep the final model prognostic factors significant at a 5% level.

## 3. Results

### 3.1. Patient and Treatment Characteristics

The patient characteristics are detailed in Table 1. From 1 January to 31 June 2016, 538 patients were included in 76 centers (Figure 1). Patients were predominantly male (*n* = 288; 53.3%) and the median age was 71.60 (63.80–78.30). Among the known risk factors of PDAC, the presence of diabetes was reported in 156 patients (29.0%) and history of smoking was reported in 173 patients (32.2%). The onset symptoms preceding cancer detection were pain, jaundice, and weight loss in 295 (54.8%), 135 (25.1%), and 121 (22.5%) patients, respectively. Diagnosis was made after an incidental finding in 57 patients (10.6%). At the time of diagnosis, 116 patients (21.8%) had a resectable disease, 64 (12.0%) had a borderline tumor, 147 (27.6%) had a locally advanced tumor, and 205 (38.5%) had a metastatic disease. Patient care included chemotherapy in 401 (74.7%) patients, surgery in 181 (33.6%) patients with 147 (27.3%) primary tumor surgical resections, radiation therapy in 44 (8.2%) patients, and exclusive best supportive care in 100 (18.6%) patients. Surgery was performed in low-volume (performing less than 5/year), medium-volume (5 to 20), or high-volume (˃20/year) centers in 21 (11.8%), 69 (38.8%), and 88 (49.4%) cases (missing data in 3 patients), respectively, and 63 (34.8%) patients experienced surgery-related adverse events. For resection surgeries, the margins were R0 and R1 in 125 (85.0%) and 22 (15.0%) patients, respectively. Patients had access to supportive care such as nutritional support in 280 (62.6%) cases and psychological follow-up in 70 (15.7%) cases, and 6.5% had a specialized care of pain.

### 3.2. Care Pathway and Delays

Patient care pathway delays are illustrated in Figure 2. In the case of symptomatic disease, the median delay between symptom onset and first CT scan was 29 days (IQR: 12.0–69.5). When the first symptom was jaundice, this delay was reduced to 14.0 (8.0–27.0) days versus 39.0 (17.0–90.0) days in non icterous patients. An endoscopic ultrasound was performed in 510 patients within a median delay of 42.0 (22.0–87.5) days after first symptoms. Diagnosis was histologically proven in 494 patients, within a median delay of 20.0 (12.0–40.0) days after CT scan. The diagnosis announcement consultation took place within a median delay of 23.5 (14.0−41.0) days after the CT scan. The therapeutic strategy was discussed in 517 cases in a multidisciplinary concertation meeting, as recommended by French practice guidelines, with a median delay of 27 (16.5–49) days after the first CT scan. When a therapeutic option different from the exclusive supportive care was possible, treatment was initiated within 43.0 (28.0–65.0) days after first imaging (*n* = 438).

### 3.3. Survival Outcomes and Predictive Markers of Survival

The median follow-up of the global cohort was 62.3 (IQR: 59.7–63.7) months, with a median OS of 9.4 (95CI: [8.3–10.9]) months, and a 12-month and 24-month OS of 42% (95CI: 37–46) and 20% (95CI: 17–23), respectively (Figure 3A). The factors associated with OS in the univariate analyses are detailed in Appendix A. After multivariate analyses, factors associated with a decreased OS were being over 70 years old (HR = 1.462, 95CI: [1.22–1.75], *p* < 0.0001), disease stage at diagnosis (HR_locally advanced_ = 1.768; 95CI: [1.38–2.26] and HR_metastatic_ = 6.11; 95CI: [4.68–7.98] versus resectable disease, *p* < 0.0001) (Figure 3B), and the presence of symptoms at diagnosis (HR = 1.564; 95CI: [1.15–2.13], *p* = 0.0042) (Table 2).

In the global cohort, delay between first symptoms and the fist CT scan of ≤ 15 days was not found to significantly improve the overall survival (*p* = 0.1091). In patients with access to a treatment, including surgery, chemotherapy, and/or radiotherapy (*n* = 438), a delay between CT scan and treatment initiation of > 21 days was associated with a significant increase in OS (HR = 0.534; 95CI: [0.44–0.65]) in the univariate (Appendix A) without significancy after multivariate analyses (HR = 0.884; 95CI: [0.66–1.19], *p* = 0.4220) (Appendix A). In the univariate analysis, a similar result was observed in metastatic patients, with an OS of 3.9 months when the delay was under 21 days (*n* = 16) versus 5.8 months in patients treated after 21 days (*n* = 115) (HR = 2.02; 95CI: [1.18–3.45], *p* = 0.0085). On the contrary, in patients considered resectable at baseline, this delay did not statistically influence survival when treatment was initiated within 21 days (*n* = 26), with an OS of 20.3 months versus 22.1 months when resection was performed beyond 21 days (*n* = 89) (HR = 0.95; 95CI: [0.58–1.56], *p* = 0.847) (Appendix A).

Regarding the socio-spatial determinants, FDep, LPA, population density, GeoClasH, or spatial accessibility did not statistically modify OS in the univariate or multivariate analyses (Appendix A) in the global cohort. In resected patients, OS increased with the volume of pancreatic cancer surgery (HR_<5 surgeries/year_ = 2.236 95CI [1.34–3.72] and HR_5-20 surgeries/year_ = 1.215; 95CI: [0.86-1.72], *p* = 0.0081 compared with centers with more than 20 pancreatic surgeries per year) (Table 3). Regarding safety, median hospitalization duration, complication rate, and intensive care unit admission after surgery were not different according to the volume of cancer surgery. Surgery-related mortality represented by the 30-day death rate after surgery was 9.5, 4.3, and 1.1% in centers with less than 5, between 5 and 20, and more than 20 pancreatic surgeries per year, respectively (*p* = 0.11) (Appendix A).

Finally, several socio-spatial determinants were significantly more often associated with resectable disease at the diagnosis, such as having good access to GP (APL ≥ 74.3; OR = 1.673, 95CI: [1.10–2.54]; *p* = 0.0153), or living in municipalities with a population density below 795.1 (HR = 1.881; 95CI: [1.20–2.94] *p* = 0.0057). Proximity to an expert center did not statistically influence chances to be resectable (Table 4).

## 4. Discussion

This regional study included 538 patients with all-stage PDAC over a short period of 6 months in the AuRA region, representing more than 10% of the French population with diverse geographic contexts and various accessibilities to healthcare. This cohort, gathering clinical information from 76 facilities of the AuRA region, offers an instantaneous overview of PDAC diagnostic and therapeutic pathways in a Western country such as France, with an in-depth characterization of patient characteristics, care delays, and socio-spatial inequalities in orderto provide knowledge about the management of PDAC patients and potential obstacles to work on, in order to improve care pathways. Characteristics of the population regarding stage, different treatment options, and survival were consistent with the literature [2,15]. After multivariate analyses, classical clinical features were associated with OS such as age, stage, and the presence of symptoms, which is often a sign of a more aggressive or advanced disease.

Regarding more structural and organizational considerations, it appeared that a delay between first symptoms and CT scan did not influence OS, and that a short delay between CT scan and first treatment was associated with a shorter OS without significance after the multivariate analyses. When focusing on the subgroup analyses, no difference was observed in resectable patients at baseline, while a delay less than 21 days was associated with decreased OS in metastatic patients. Multivariate analyses were not performed on subgroups due to their small size. These results point out the fact that the intuitive idea of a shorter delay being of benefit to patients in terms of survival is probably wrong in PDAC. Moreover, this hypothesis is controversial, with many heterogeneous delay definitions and cut-offs in the literature. Raptis et al. showed that tumor resectability and time from first symptoms to referral were the only factors significantly impacting the resectability rate and survival in 335 patients with localized PDAC [5]. However, Sanjeevi et al. observed that a delay between the first imaging and resection > 32 days was associated with an increased risk of progress from resectable to unresectable in the univariate analysis without significance after multivariate in 349 patients [7]. In a series of 168 patients treated with upfront surgery for localized PDAC, Brugel et al. found that the delay between the first specialist consultation and surgery had no significant impact on DFS, OS, or 90-day morbidity [6]. Based on these studies, tumor size and degree of local involvement seem to be the strongest factors associated with survival, even though surgical retrospective series are biased. Indeed, they usually do not integrate patients who already became unresectable before data collection due to a too long delay of care, and they may exclude rapidly progressive diseases. Moreover, the study of delays in retrospective studies, such as in PANDAURA, may induce other biases such as the fact that aggressive diseases associated with symptoms and general status deterioration are more rapidly diagnosed, leading to an acceleration of patient care and thus a paradoxical result, linking short delay of care with shorter survival. Even though the question of the impact of delay on PDAC prognosis remains unresolved, it appears that one of the biggest challenges in pancreatic cancer is to detect it early enough so as to increase its chances to be resectable.

Several series have intended to show the impact of social, spatial, and racial disparities on pancreatic cancer patient outcomes. These disparities can be explained by the later detection and diagnosis in these communities, related to lower access to GPs, but also to the lower accessibility and quality of cancer management [16,17,18,19,20]. In PANDAURA, socio-economic determinants did not statistically influence the survival of PDAC patients, but several geographic inequalities in access to care were observed, with a significant impact on patient outcomes. Indeed, patients with better spatial accessibility to GP (APL ≥ 74.3; OR = 1.673, *p* = 0.0153) and living in municipalities with a population density below 795.1 people/km^2^ (HR = 1.881, *p* = 0.0057) were associated with better chances to be resectable at diagnosis. These data support the idea that the early detection of pancreatic cancer is mostly dependent on an efficient primary care network nearby, and confirms the pivotal role of GP in cancer detection programs [21,22]. The distance to an expert center did not statistically impact the stage at diagnosis nor the survival outcomes, as previously described in this pathology [23]. Nonetheless, in the case of surgery, the level of activity for the healthcare center was a determining fctor, as the operating volume was correlated with resected patient survival (HR_<5 surgeries/year_ = 2.236, HR_5-20 surgeries/year_ = 1.215, *p* = 0.0081). Surgery safety was not statistically different between center categories based on their surgical volume. However, the30-day mortality rate of 9.5% observed in centers with less than five surgeries per year, and the 30-day mortality rate of 1.1% in high-volume centers (*p* = 0.11) may suggest that surgery-related risk is also influenced by the surgical expertise of healthcare centers. These results are consistent with previous studies and support the fact that pancreatic surgeries should be centralized in high-volume centers to decrease surgery-related morbidity and mortality, as well as to improve cancer-related survival outcomes [24,25,26].

As mentioned above, this study has some limitations, mostly because of its retrospective character, but it is the largest European cohort study focusing on care pathways, as well as the socioeconomic and geographic determinants affecting pancreatic cancer patient survival. Finally, the results of this study opened the way to the second phase of PANDAURA program: a multicentric prospective public health initiative aiming to structure and harmonize the PDAC patient healthcare pathway to reduce socio-spatial disparities between patients and to optimize the management of pancreatic cancer.

## 5. Conclusions

This cohort study helped to determine key individual and collective factors influencing pancreatic cancer patient outcomes, especially in those eligible for curative surgery. Easy access to GPs was associated with better chances to be resectable at the diagnosis, and a resection in a high-volume center was associated with improved survival outcomes. This kind of cohort study is essential in order to identify the potential advantages that may allow for significant improvements in cancer patient care, but also to measure routine-setting quality of care. New public healthcare strategies should systematically integrate the pivotal role of GP in cancer and the importance of centralized pancreatic surgeries in expert centers to optimize PDAC patient care pathways and outcomes.

## Figures and Tables

**Figure 1 cancers-14-05413-f001:**
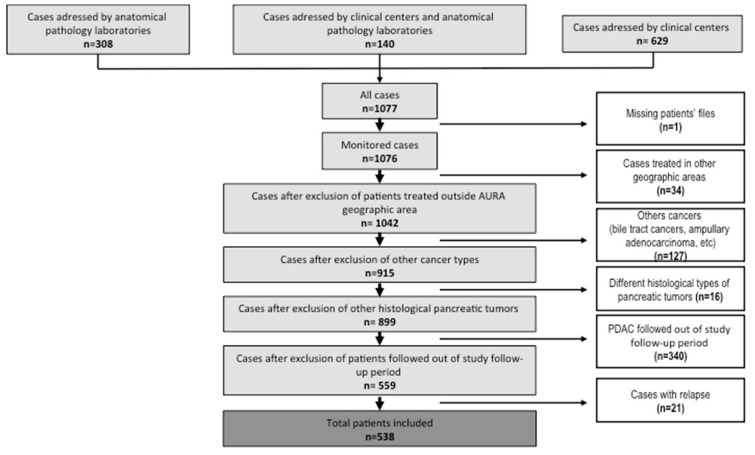
Flowchart.

**Figure 2 cancers-14-05413-f002:**
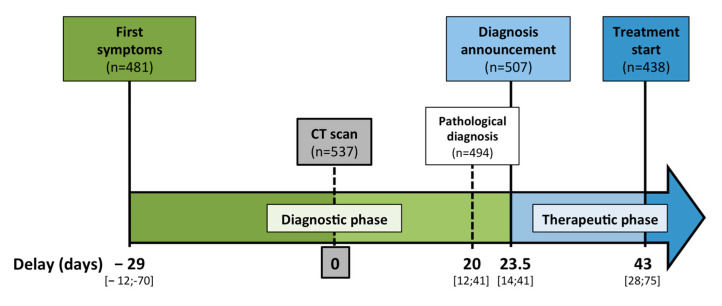
Timeline of the patient care pathway with median delays for each diagnostic and therapeutic step. As only 481 patients presented symptoms before the diagnosis, delay between symptoms and CT scan did not integrate patients with incidental disease. From the CT scan, the whole population was considered to determine the timeline with median delays. Treatment start excluded patients with exclusive best supportive care, as this care usually started without delay if needed. Delays are presented as median with IQR.

**Figure 3 cancers-14-05413-f003:**
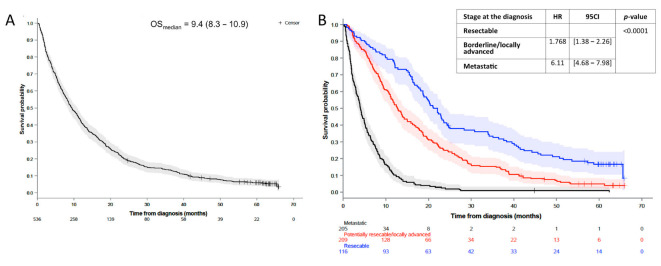
Overall survival of (**A**) the global cohort and (**B**) according to different stages at the diagnosis, as follows: resectable, borderline or locally advanced disease, and metastatic disease. Overall survival is presented as a median with a 95% confidence interval. The results were figured using the Kaplan–Meier method. Numbers at risk are reported under the x-axis. HR presented in the figure (**B**) are obtained after the comparison of OS between the groups after the multivariate analysis. HR: hazard ratio; OS: overall survival; 95CI: 95% confidence interval.

**Table 1 cancers-14-05413-t001:** Patient and treatment characteristics.

Patients’ Characteristics	All	
	*n* = 538	
**Clinical and biological characteristics**		
**Sex**		
Male	288	(53.5%)
Female	250	(46.5%)
**Age ***	71.60 [63.80–78.30]	
≤70	238	(44.2%)
>70	300	(55.8%)
**BMI**		
<20	58	(10.8%)
[20–24]	243	(45.2%)
[25–29]	161	(29.9%)
≥30	76	(14.1%)
**Past medical history**		
Diabetes	156	(29.0%)
Tobacco	173	(32.2%)
Previous cancer	79	(14.7%)
**Circumstances of cancer diagnosis**		
Incidental	57	(10.6%)
Presence of symptoms	481	(89.4%)
Abdominal pain	295	(54.8%)
Jaundice	135	(25.1%)
Loss of weight	121	(22.5%)
Other symptoms	173	(32.2%)
**Stage at the diagnosis**		
Missing data	6	
Resectable	116	(21.8%)
Borderline	64	(12.0%)
Locally advanced	147	(27.6%)
Metastatic	205	(38.5%)
**Histological differenciation grade**		
Missing data	233	
Well differentiated	77	(25.2%)
Moderately differentiated	178	(58.4%)
Poorly differentiated	50	(16.4%)
**Socio-spatial determinants**		
**French deprivation index (Fdep) ***	−0.11 [−1.05–0.80]	
≤−1.3	105	(21.4%)
[−1.2; −0.5]	115	(20.8%)
[−0.4; 0.3]	112	(18.2%)
[0.4; 1]	98	(20.1%)
>1	108	(19.5%)
**Accessibility of General Practitioners** **(Localized Potential Accessibility index) ***	70.43 [52.74–82.61]	
≤46.7	108	(19.7%)
[46.8; 63.8]	106	(21.9%)
[63.9; 74.3]	118	(18.4%)
[74.4; 86.4]	99	(19.9%)
>86.4	107	(20.1%)
**Population density * (** **inhabitants/km^2^)**	446.55 [107.62–2132.50]	
≤83.7	107	(19.9%)
[83.8; 254.7]	109	(20.3%)
[254.8; 798.1]	108	(20.1%)
[798.2; 2159.9]	106	(19.7%)
>2159.9	108	(20.1%)
**Travel time to the nearest specialized center (in minutes)**	48.0 [18.0–71.0]	
≤13	115	(21.4%)
[14; 36]	101	(18.8%)
[37; 58]	109	(20.3%)
[59; 79]	107	(19.9%)
>79	106	(19.7%)
**GeoClasH Classification**		
Wealthy Metropolitan Areas	139	(25.8%)
Precarious Population Districts	249	(46.3%)
Residential Outskirts	80	(14.9%)
Agricultural and Industrial Plains	17	(3.2%)
Rural Margins	53	(9.9%)
**Therapeutic strategies**		
Curative intent		
Surgical resection alone	26	(4.8%)
Surgical resection with chemotherapy	108	(20.1%)
Surgical resection with chemotherapy and radiation therapy	13	(2.4%)
Palliative intent		
Chemotherapy alone	230	(42.8%)
Chemotherapy with radiation therapy	24	(4.5%)
Radiation therapy alone	3	(0.6%)
Other types of surgery alone	8	(1.5%)
Other types of surgery with chemotherapy	22	(4.1%)
Other types of surgery with chemotherapy and radiation therapy	4	(0.7%)
Exclusive best supportive care	100	(18.6%)

Data marked with * are represented by median with [Q1–Q3]. BMI = body mass index.

**Table 2 cancers-14-05413-t002:** Impact of clinical features and socio-spatial determinants on the overall survival in the global cohort after multivariate analyses. The final model included 530 patients. The number presented between parentheses in the *p*-value column corresponds to the rank of the variable in the multivariate analysis based on its *p*-value. *p*-values < 0.05 are marked in bold.

Variables	Value	HR	IC95	*p*-Value
**Clinical characteristics**				
**Sex**	Female			NS(1)
	Male			
**Age (years)**	≤70			**<0.0001**
	>70	1.462	[1.22–1.75]	
**BMI**	<20			NS(4)
	[20–24]			
	[25–29]			
	≥30			
**Presence of diabetes**	No			NS(7)
	Yes			
**Circumstances of cancer diagnosis**	Incidental diagnosis			**0.0042**
	Presence of symptoms	1.564	[1.15–2.13]	
**Stage at the diagnosis**	Resectable			**<0.0001**
	Borderline/locally advanced	1.768	[1.38–2.26]	
	Metastatic	6.11	[4.68–7.98]	
**Socio-spatial determinants**				
**French deprivation index**	≤−1.3			NS(8)
	[−1.2; −0.5]			
	[−0.4; 0.3]			
	[0.4; 1]			
	>1			
**Accessibility of General Practitioners** **(Localized Potential Accessibility index)**	≤46.7			NS(10)
	[46.8; 63.8]			
	[63.9; 74.3]			
	[74.4; 86.4]			
	>86.4			
**Population density** **(inhabitants/km^2^)**	≤83.7			NS(3)
	[83.8; 254.7]			
	[254.8; 798.1]			
	[798.2; 2159.9]			
	>2159.9			
**GeoClasH Classification**	Wealthy Metropolitan Areas areas			NS(6)
	Precarious Population Districts areas			
	Residential Outskirts			
	Agricultural and Industrial Plains			
	Rural Margins			
**Travel time to the nearest specialized center (in minutes)**	≤13			NS(5)
	[14; 36]			
	[37; 58]			
	[59; 79]			
	>79			

BMI = body mass index; NS = non-significant.

**Table 3 cancers-14-05413-t003:** Impact of clinical features and socio-spatial determinants on overall survival in resected patients after multivariate analyses. The final model included 177 patients. The number presented between parentheses in the *p*-value column corresponds to the rank of the variable (rank) in the multivariate analysis based on its *p*-value. *p*-values < 0.05 are marked in bold characters.

Variables		HR	IC95	*p*-Value
**Clinical characteristics**				
**Sex**	Female			NS(5)
	Male			
**Age class**	≤70			NS(6)
	>70			
**BMI**	<20			NS(10)
	[20–24]			
	[25–29]			
	≥30			
**Presence of diabetes**	No			NS(9)
	Yes			
**Circumstances of cancer diagnosis**	Incidental diagnosis			NS(11)
	Presence of symptoms			
**Stage at the diagnosis**	Resectable			NS(4)
	Borderline/locally advanced			
	Metastatic			
**Surgical features**				
**Resection margin status**	R0			NS(8)
	R1			
**Surgery-related adverse events**	No			NS(7)
	Yes			
**Number of pancreatic surgeries per center per year**	<5	2.236	[1.34–3.72]	**0.0081**
	5–20	1.215	[0.86–1.72]	
	>20			
**Socio-spatial determinants**				
**French deprivation index**	≤−1.3	1.336	[0.80–2.24]	**0.0199**
	[−1.2; −0.5]	0.575	[0.34–0.98]	
	[−0.4; 0.3]			
	[0.4; 1]	1.136	[0.67–1.92]	
	>1	0.888	[0.52–1.50]	
**Accessibility of General Practitioners** **(Localized Potential Accessibility index)**	≤46.7			NS(12)
	[46.8; 63.8]			
	[63.9; 74.3]			
	[74.4; 86.4]			
	>86.4			
**Population density (inhabitants/km^2^)**	≤83.7			NS(3)
	[83.8; 254.7]			
	[254.8; 798.1]			
	[798.2; 2159.9]			
	>2159.9			
**GeoClasH Classification**	Wealthy Metropolitan Areas			NS(2)
	Precarious Population Districts			
	Residential Outskirts			
	Agricultural and Industrial Plains			
	Rural Margins			
**Travel time to the nearest specialized center (in minutes)**	≤13			NS(1)
	[14; 36]			
	[37; 58]			
	[59; 79]			
	>79			

BMI = body mass index; NS = non-significant.

**Table 4 cancers-14-05413-t004:** Impact of socio-spatial determinants on disease stage at the diagnosis in the global cohort after univariate analysis. *p*-values < 0.05 are marked in bold characters.

Variables		OR	IC 95	*p*-Value
**Accessibility of General** **Practitioners**	<74.3			**0.0153**
	≥74.3	1.673	[1.10–2.54]	
**Population density** **(inhabitants/km^2^)**	<795.1	1.881	[1.20–2.94]	**0.0057**
	≥795.1			
**Travel time to the nearest** **specialized center (in minutes)**	<58			0.2162
	≥58	1.300	[0.86–1.97]	

## Data Availability

Data presented in this study are available upon request to the corresponding author. The data are not publicly available due to confidential data among the data sets.

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
