# Peer review of "Structural and Socio-Spatial Determinants Influencing Care and Survival of Patients with a Pancreatic Adenocarcinoma: Results of the PANDAURA Cohort"

_cancers, 2022, doi:10.3390/cancers14215413_

Round 1

Reviewer 1 Report

This study is a study of patients in the 'AURA' (Auvergne-Rhône-Alpes) area, and I think the contents may be different about the medical facilities and accessibility of the area. Therefore, it would have been better if there was an explanation of the medical facilities in the area.

If it's a medical facility in a city, I wonder if other data might have had an impact. I hope that there is a comparison with the results of studies in other cities or countries to verify more objectivity.

Author Response

Reviewer 1 :

This study is a study of patients in the 'AURA' (Auvergne-Rhône-Alpes) area, and I think the contents may be different about the medical facilities and accessibility of the area. Therefore, it would have been better if there was an explanation of the medical facilities in the area.

If it's a medical facility in a city, I wonder if other data might have had an impact. I hope that there is a comparison with the results of studies in other cities or countries to verify more objectivity.

Answer : We thank reviewer 1 for his/her comments. Introduction has been modified as follow to give more details on:

« Thus, the French regional Pancreas Program PANDAURA, launched in 2019 aimed to analyze care pathways of PDAC patients and provide useful knowledge to improve them and reduce potential inequalities in Auvergne-Rhône-Alpes (AuRA), which is a large region (69 711 km2) of 8,000,000 inhabitants (approximately 12% of the French population) with very heterogenous territories such as metropoles, agricultural areas, or mountains. This project brought together various key healthcare players in AuRA as the Regional Health Agency, the Regional Cancer Network, some regional unions of health care practitioners (GP, nurses, specialties), as well as a broad spectrum of clinical centers such as Universitary Hospitals, Comprehensive Cancer Centers,  health care facilities from public (Hospital) and private(Clinics) sectors.»

Reviewer 2 Report

Please see attached word document

Author Response

Reviewer 2 :

This study included a large cohort of patients identified from pathology reports and pancreatic cancer multidisciplinary team meetings in a French region comprising 8 million inhabitants, thus identifying 538 patients with PDAC diagnosed during a period of 6 months in 2016. The manuscript is written in good language, is well-structured and has novel findings with interest to the readers of Cancer and the special issue on “Gastrointestinal Tract Cancers, an Increasing Burden of the Modern Era: Epidemiology and Prevention”. The main limitation lies in the retrospective design, for example with respect to possible inclusion bias and heterogeneity, and furthermore, the novelty lies not so much in analysis of primary objectives but mostly in secondary analyses/objectives and completeness of data presentation.

The beauty of the study is the structured data presentation, and the study may inspire further care-path improvements, although interpretation must be done with some caution.

There is need for some corrections in English language (especially in the “Simple summary” paragraph, but also for example Table 1, “Cancer antérieur” should be written “Previous cancer”; more examples are listed below). The main impression is however that this manuscript may be accepted upon minor revision.

To give a more detailed description and criticism of the manuscript, the primary objective was to identify medical (clinical?), environmental (or rather geographic?) and social determinants of PDAC patients’ care and overall survival. The main findings, pertaining to the primary objectives, were

-          (with respect to patients’ care, as shown in Figure 2:) that there was a rather long time span from first symptoms to diagnostic CT scan (interquartile range 12 to 70 days) and from diagnostic CT to treatment (interquartile range 28 to 75 weeks), and

-          (with respect to overall survival, as shown in Figure 3 and Table 2:) that there was a significant association between overall survival for the whole cohort and high age, resectability, and whether the disease was symptomatic.

These findings with respect to the primary objectives were as expected and perhaps not so novel, except that the study beautifully presents real-world descriptive data and thus gives an interesting and truthful insight into the care pathway in a representative, large region with a developed health system. The novelty thus lies primarily in the secondary analyses.

Pertaining to secondary objectives, the study showed that (Table 3) pancreatic surgery institutional volume and French deprivation index (measuring income, education, unemployment and working-class jobs) were independent predictors of survival in subgroup analysis of resected patients (n=177). I also don’t believe this to be novel, but it does confirm that socio-economic patient factors and high-specialized treatment factors contribute to the disparities we seen in care and survival for PDAC. Interestingly, (Table 4), univariate analysis of the whole cohort showed that low accessibility to general practitioner and high population density were both associated with poor survival. These (secondary) findings point to possible actionable structural deficiencies, although again, the retrospective nature of the study precludes definitive judgement.

The authors should be commended for clarity in data presentation. They could consider adding the two following two additional pieces of data (to the text of the manuscript):

-          On page 4, where there is a description of surgery and adverse events in centres with different volume: please add peri-operative (30-day) death for the separate categories of pancreatic surgery institutional volume (<5 vs 5-20 vs >20 pancreatic surgeries per year)

Answer:

We thank reviewer 2 for this remark. Informations have been added in results, discussion and supplementary table 3 as follows:

Results

Regarding safety, median hospitalization duration, complication rate and intensive care unit admission after surgery were not different according to the volume of cancer surgery. Surgery-related mortality represented by 30-day death rate after surgery was respectively 9.5, 4.3, 1.1% in centers with less than 5, between 5 and 20 and more than 20 pancreatic surgeries per year (p=0.11) (Supplementary table 3)

Discussion

Surgery safety was not statistically different between centers categories based on their surgical volume. However, the 30-days mortality rate of 9.5% observed in centers with less than 5 surgeries per year while 30-day mortality rate is 1.1% in high-volume centers (p=0.11) may suggest that surgery-related risk is also influenced by surgical expertise of healthcare centers. These results are consistent with previous studies and support the fact that pancreatic surgeries should be centralized in high-volume centers to decrease surgery-related morbidity and mortality, as well as improve cancer-related survival outcomes [28,30].

Supplementary table 3: Surgery-related adverse events and mortality according to volume of pancreatic surgery per center

Number of surgeries per center per year

All

Test

Missing data

5-20

<5

>20

N=3

N=69

N=21

N=88

N=181

Surgery related adverse events ?

Chi-2
   P = 0.927

   Missing data

0

1

0

0

1

   No

2

(66.7%)

43

(63.2%)

14

(66.7%)

58

(65.9%)

117

(65.0%)

   Yes

1

(33.3%)

25

(36.8%)

7

(33.3%)

30

(34.1%)

63

(35.0%)

Duration of hospitalization (days)

Kruskal-Wallis
   P = 0.657

   N

3

69

21

88

181

   Mean (SD)

19.67 (7.51)

18.29 (15.66)

16.10 (12.08)

16.32 (8.76)

17.10 (12.15)

   Median (min; max)

20.00 (12.00; 27.00)

14.00 (3.00; 116.00)

13.00 (3.00; 58.00)

15.00 (2.00; 58.00)

14.00 (2.00; 116.00)

   IQR [Q1 - Q3]

[12.00 - 27.00]

[10.00 - 20.00]

[10.00 - 17.00]

[10.00 - 19.00]

[10.00 - 20.00]

Transfer in intensive care unit

Chi-2
   P = 0.393

   No

2

(66.7%)

57

(82.6%)

16

(76.2%)

77

(87.5%)

152

(84.0%)

   Yes

1

(33.3%)

12

(17.4%)

5

(23.8%)

11

(12.5%)

29

(16.0%)

Death within 30 days after surgery

0

(0.0%)

3

(4.3%)

2

(9.5%)

1

(1.1%)

6

(3.3%)

 Fisher test

P = 0.11

-          On page 8, in the paragraph beginning with “In the global cohort, delay between first symptoms and the first CT scan did not influence survival.”

o          Please add a relevant comparison, for example by splitting the cohort by the median time from first symptoms to CT scan, and add the corresponding p-value for this comparison of lead-time vs survival.

Answer:

We thank reviewer 2 for this remark, the sentences has been modified with precisions on the analysis as follows: « In the global cohort, delay between first symptoms and the fist CT scan ≤15 days was not found to significantly improve overall survival (p=0.29). »

Furthermore, I am curious whether the significance of lead-time from symptoms to CT could be more important for poorly differentiated tumours; i.e., were poorly differentiated tumours less often subject to curative-intent resection when there was a high (vs low) lead-time from symptoms to CT? [Time to CT may be more relevant in this respect than time to resection, because CT objectively defines resectability.] Consequently, the chance of being offered a curative-intent resection and obtain long-term survival might be associated with the histologic grade, and a multivariable survival analysis with added interaction term for histological grade vs lead time to CT might show such an interaction.

Therefore, as a supplementary table, please add a subgroup analysis showing lead-time vs histological grade as outlined below. Please also add an interaction analysis to see whether a possible association between lead-time to CT and survival could depend on the histological grade. For this analysis, I would suggest grouping by differentiation into “well or moderate” vs “poor”.

A

B

C

D

1

Histological grade

Lead time from symptoms to CT

N

HR and CI

p-value

2

Well differentiated

less than median lead-time, range

n=?

reference

p (C3 vs C2)

3

more than median lead-time, range

n=?

[C3]

4

Moderately differentiated

less than median lead-time, range

n=?

reference

p (C5 vs C4)

5

more than median lead-time, range

n=?

[C5]

6

Poorly differentiated

less than median lead-time, range

n=?

reference

p (C7 vs C6)

7

more than median lead-time, range

n=?

[C7]

Interaction analysis for histological grade vs lead time to CT with respect to survival

Histological grade

Well or moderately differentiated

Poorly differentiated

Lead-time to CT

Less than median

More than median

Interaction term

Grade vs Lead time

Answer:

We thank reviewer 2 for these suggestions. As presented below, median delay is not impacted by differentiation grade and due to small number of patients, we cannot observed differences of survival according to delay and differentiation degree as shown below:

Histological grade

All

Donnée manquante

G1. Well differentiated

G2. moderately differentiated

G3. Poorly differentiated

N=233

N=77

N=178

N=50

N=538

Lead-time to CT

   N

209

69

156

46

480

   Mean (SD)

58.10 (88.80)

43.54 (55.87)

51.99 (59.10)

54.39 (63.95)

53.66 (73.53)

   Median (min; max)

31.00 (0.00; 948.00)

21.00 (0.00; 264.00)

28.00 (0.00; 325.00)

28.00 (1.00; 339.00)

29.00 (0.00; 948.00)

   IQR [Q1 - Q3]

[13.00 - 69.00]

[9.00 - 49.00]

[15.00 - 63.00]

[13.00 - 91.00]

[12.00 - 69.50]

Lead-time to CT

   Missing data

24

8

22

4

58

   <=15 days

65

(31.1%)

29

(42.0%)

42

(26.9%)

15

(32.6%)

151

(31.5%)

   >15 days

144

(68.9%)

40

(58.0%)

114

(73.1%)

31

(67.4%)

329

(68.5%)

Lead-time to CT

   Missing data

24

8

22

4

58

   <=29 days

101

(48.3%)

38

(55.1%)

80

(51.3%)

24

(52.2%)

243

(50.6%)

   >29 days

108

(51.7%)

31

(44.9%)

76

(48.7%)

22

(47.8%)

237

(49.4%)

Finally, here is a list of language/typographical errors identified during this review:

-          Simple summary, line 2: “and a highly lethal” should be “and highly lethal”. “Optimize its diagnosis” should be “Optimizing its diagnosis”. Line 4, “Among most important results” should be “Among the most important results”.

-          Patients and method, Population, line 1, “Inclusion criteria were as follow” should be “… as follows”. Line 5, “others histological diagnoses” should be “other histological diagnoses”. Line 6, “every health care institutions” should be “every health care institution”. Line 8, “useful clinical informations were collected” should be “useful clinical information was collected”.

-          Socio-spacial indices: line 4: “from a set of four variables measuring (income, education, unemployment and working-class jobs)” should be “from a set of four variables (measuring income, education, unemployment and working-class jobs)”

-          Statistics, last line, “prognostic factors significant a 5% level” should be “prognostic factors significant at a 5% level”

-          Results, Patients’ and treatment characteristics paragraph, line 10, “primitive tumor surgical resections” should be “primary tumor surgical resections”

-          All tables: left brackets should be [

-          Figure 1, box containing “Cases after exclusion of patients followed cas out of study follow-up period” should read “Cases after exclusion of patients followed out of study follow-up period”

-          Figure 2, legend, “does not integrate patients with indental disease” should be “… with incidental disease”

-          Figure 3 should be in higher resolution, since the text renders blurry in the pdf version presented to me

-          Page 8, paragraph beginning with “In the global cohort, …”, an ‘r’ is missing in ‘first’: “… and the fist CT scan”

-          Discussion, second paragraph, missing punctuation before and after the word “Besides” in line 25. This should read “… may exclude rapidly progressive diseases. Besides, study of delays in …”

-          Discussion, last paragraph, “As above-mentionned,” should be “As mentioned above,”

Answer: we thank you reviewer 2 for these corrections. Modifications have been done.

Reviewer 3 Report

The manuscript titled “Structural and socio-spatial determinants influencing care and survival of patients with a pancreatic adenocarcinoma: results of the PANDAURA cohort” describes the important impact of treatment and general practitioner in cancer care on the outcomes of pancreatic cancer patients. The followings are some concerns and comments have been pointed out that the authors may want to consider.

1) I’d suggest the authors provide a version with line numbers for easier tracking.

2) Please use italic p as it refers to a p-value throughout the manuscript.

3) The “general practitioner” only appears one time in the main content. Please consider other more suitable keywords if the authors do not mind. Please check other keywords as well.

4) I’d highly suggest the authors extend the introduction section.

5) Please provide higher resolution figures.

6) Please be consistent with the capital letter or lowercase “n” throughout the manuscript.

7) Please extend the Figure 3 legend and include brief statistical information.

8) Table 2, Table 3: Some of the brackets are left-side-right or right-side-left. Please correct them and check throughout the manuscript.

9) Discussion section, paragraph 2 is too long and is not easy to follow.

10) Please enrich discuss section with more information and references.

Author Response

Reviewer 3 :

The manuscript titled “Structural and socio-spatial determinants influencing care and survival of patients with a pancreatic adenocarcinoma: results of the PANDAURA cohort” describes the important impact of treatment and general practitioner in cancer care on the outcomes of pancreatic cancer patients. The followings are some concerns and comments have been pointed out that the authors may want to consider.

We thank the reviewer 3 for his/her comments. Please find ourpoint by point answers below:

1) I’d suggest the authors provide a version with line numbers for easier tracking.

Answer: This problem has been adressed

2) Please use italic p as it refers to a p-value throughout the manuscript.

Answer: This problem has been adressed

3) The “general practitioner” only appears one time in the main content. Please consider other more suitable keywords if the authors do not mind. Please check other keywords as well.

Answer: This keyword has been removed

4) I’d highly suggest the authors extend the introduction section.

Answer: Details on AuRA specificities have been added to justify the relevance of the cohort as follows :

« Thus, the French regional Pancreas Program PANDAURA, launched in 2019 aimed to analyze care pathways of PDAC patients and provide useful knowledge to improve them and reduce potential inequalities in Auvergne-Rhône-Alpes (AuRA), which is a large region (69 711 km2) of 8,000,000 inhabitants (approximately 12% of the French population) with very heterogenous territories such as metropoles, agricultural areas, or mountains. This project brought together various key healthcare players in AuRA as the Regional Health Agency, the Regional Cancer Network, some regional unions of health care practitioners (GP, nurses, specialties), as well as a broad spectrum of clinical centers such as Universitary Hospitals, Comprehensive Cancer Centers,  health care facilities from public (Hospital) and private(Clinics) sectors»

5) Please provide higher resolution figures.

Answer: High quality figures are up load separately from main document

6) Please be consistent with the capital letter or lowercase “n” throughout the manuscript.

Answer: This problem has been adressed

7) Please extend the Figure 3 legend and include brief statistical information.

Answer: This problem has been adressed. Legend has been modified as follows: « Overall survival of (A) the global cohort and (B) according to different stages at the diagnosis as follow: resectable, borderline or locally advanced disease, and metastatic disease. Overall survival is presented as a median with 95% confidence interval.  The results were figured using Kaplan-Meier Method. Numbers at risk were reported under the x-axis HR presented in figure B were obtained after comparison of OS between groups after multivariate analysis. Abbreviations: HR: hazard ratio; OS: overall survival; 95CI: 95% confidence interval »

8) Table 2, Table 3: Some of the brackets are left-side-right or right-side-left. Please correct them and check throughout the manuscript.

Answer: This problem has been adressed

9) Discussion section, paragraph 2 is too long and is not easy to follow.

Answer: This problem has been adressed with shortening of description of surgical retrospective series studying the impact of delay on resection rate and survival.

10) Please enrich discuss section with more information and references.

Answer: This problem has been adressed with the addition of the following paragraph and references:

Several series intended to show the impact of social , spatial and racial disparities on pancreatic cancer patients outcomes. These disparities can be explained by later de-tection and diagnosis in these communities, related to lower access to GP, but also by lower accessibility and quality of cancer management. [20-24].

Round 2

Reviewer 3 Report

Thank you for the updates. Please make some minor modifications before publication.

1) Page 2 line 1 and throughout the manuscript: Please use italic p.

2) The figures are OK but not clearer when zooming in. Please provide a higher resolution by increasing the pixels.

Author Response

1) Page 2 line 1 and throughout the manuscript: Please use italic p.

2) The figures are OK but not clearer when zooming in. Please provide a higher resolution by increasing the pixels.

Problems 1)  and 2) have been addressed